# CFDiffusion: Controllable Foreground Relighting in Image Compositing via Diffusion Model

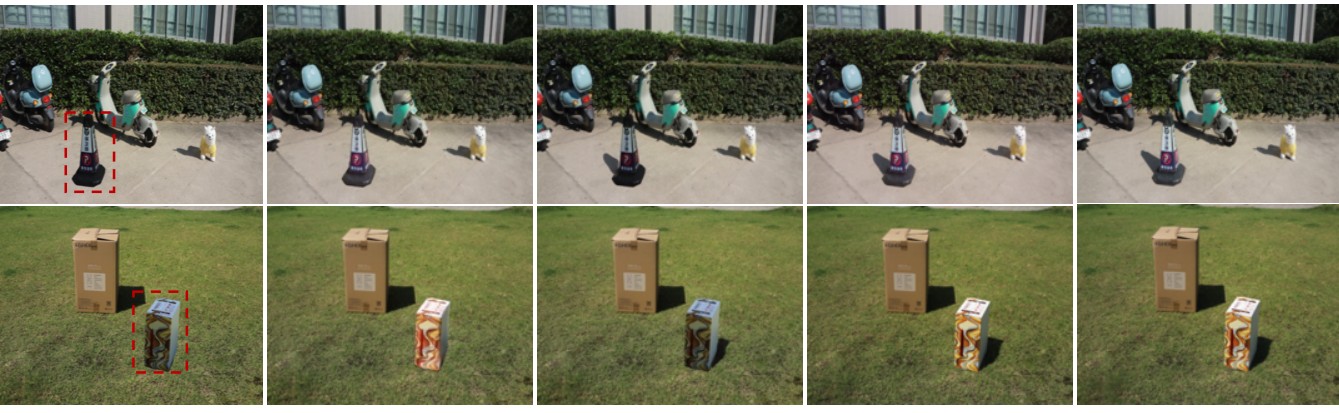

**Figure 1: In this illustration, we demonstrate the specific effects of image harmonization and shadow generation. The areas enclosed by the red rectangles represent our foreground objects. From the composite image to the ground truth (GT), we integrate these two operations into a unified model: adjusting the appearance of the foreground object to harmonize it with the background and generating reasonable shadows for the foreground object.**

## ABSTRACT

Inserting foreground objects into specific background scenes and eliminating the illumination inconsistency (eg., color, brightness) between them is an important and challenging task. It typically involves multiple processing tasks, such as image harmonization and shadow generation. In these two domains, there are already many mature solutions, but they often only focus on one of the tasks. Recently, some image composition methods have utilized diffusion models to address both of these issues simultaneously, but they cannot guarantee the complete reconstruction of foreground content. In this work, we propose CFDiffusion, which can simultaneously handle image harmonization and shadow generation. We first employ a shadow mask predictor to estimate the shadow mask of the foreground object. Next, we design a harmonization-shadow generator based on a diffusion model to harmonize the foreground and generate shadows concurrently. Additionally, we propose a foreground content enhancement module to ensure the complete preservation of foreground content at the insertion location, and we also develpp an adaptive encoder to guide the harmonization process in the foreground area. The experimental results on the iHarmony4 dataset and our IH-SG dataset demonstrate the superiority of our CFDiffusion approach.

## KEYWORDS

CFDiffusion, image composition, object relighting, image harmonization, shadow generation

## 1 INTRODUCTION

Image composition stands as a fundamental task in computer vision and augmented reality, aimed at seamlessly integrating objects from one image into another to craft a convincingly realistic composite image. Merely inserting the foreground object into a background image without careful consideration results in noticeable discrepancies between the foreground and background, such as differences in color, brightness, and shadows. Based on this, we can decompose the image compositing process into multiple subtasks, each addressing specific issues: image harmonization [2, 5, 6, 9, 19, 23, 48, 51] and shadow generation [17, 28, 44, 55].

Image harmonization aims to adjust the compatibility between the foreground and background in terms of color and brightness, while shadow generation ensures that the inserted foreground objects cast realistic and reasonable shadows. Various practical methods exist for both of these subtasks. However, employing multiple models to address each subproblem individually is both cumbersome and impractical. What we need is a unified network that can address both of these issues simultaneously, achieving excellent results for each. Figure 1 illustrates the problems we need to deal with and the results we should achieve

In recent years, generative models such as GANs [3, 18, 20, 43] and diffusion models [1, 16, 30, 34, 36, 39, 40] have demonstrated significant potential in image composition. Particularly, diffusion models have surpassed various preceding methods in image editing [1, 22, 34] and other applications [15, 33, 37]. Conditional diffusion model aims to generate images under the guidance of conditional information, such as text or semantic masks. Among them, Stable

Diffusion (SD) [39] stands out as one of the most popular models, successfully integrating text from CLIP [38] into latent diffusion.

Some existing works have introduced the diffusion model into related research domains, particularly in image editing and image composition. For example, SDEdit [34] composites images by adding noise to the input image and then iteratively denoises it through stochastic differential equations. However, these methods lacks proper and sufficient guidance during the denoising process, resulting in the final image lacking sufficient content fedelity. Besides, most diffusion models for image editing focus on manipulating images using text input, which is inappropriate for image composition.

Recently, some image compositing methods [46, 52] have attempted to address all issues within a unified model, which can significantly simplify both model size and complexity. For example, ObjectStitch [46] utilizes a bounding box that encompasses the foreground object to specify the region for foreground object insertion and shadow generation, then processes the foreground within this designated area. Given the recent successful applications of diffusion models in image processing, these methods typically rely on pretrained diffusion models. However, in practice, these methods result in uncontrollable adjustments to the foreground in terms of both position and content texture, raising concerns about preserving the fidelity and credibility of the foreground object.

In this paper, our aim is to address the issue of inadequate foreground fidelity observed in previous discussions on image composition. We introduce a method called CFDiffusion that concurrently handles image harmonization and shadow generation tasks. Building upon stable diffusion, we introduce a foreground content enhancement module (FCEM), which utilizes a foreground content encoder to extract foreground content information, thus guiding the reconstruction of foreground content. Furthermore, we equip SD with a lightweight adaptive encoder designed to extract crucial conditional information from the composite image, such as background style and color, to guide the denoising process of SD.

To validate the effectiveness of our approach, we compare it with state-of-the-art methods and conduct experiments on benchmark datasets such as iHarmony4 [7] and our proposed IH-SG dataset. The experimental results demonstrate that our method achieves more realistic image harmonization and produces shadows that are both genuine and believable.

Our contributions can be summarized as follows:

- We introduce a novel image composition method called CFDiffusion. This method simultaneously handles image harmonization and shadow generation tasks for foreground objects with masked insertion points.
- We design a foreground content enhancement module to fully reconstruct the foreground content and texture details.

Extensive experiments conducted on both public datasets and our newly created dataset IH-SG validate the effectiveness of our proposed method.

## 2 RELATED WORK

### 2.1 Image Harmonization

As a subtask of image compositing, the objective of image harmonization is to integrate objects from a given foreground image into a background image to create a cohesive composite image. This process involves adjusting the color and lighting information of the foreground object to ensure its compatibility with the background of the composite image.

Traditional methods [26, 47] rely on adjusting the appearance of the foreground to match the color statistics of the background, typically focusing on obtaining color statistics and then transferring this information between the foreground and background. These methods are fast and straightforward but often struggle with complex scenes and produce artifacts because the realism of the image is often not well captured by these statistics.

Particularly, with the release of the first large-scale image harmonization dataset, iHarmony4 [7], supervised image harmonization methods [4, 8, 10, 11, 13, 27] have garnered increasing attention. For instance, [11] employed attention blocks to compute non-local information for foreground adjustment. SSAM [8] integrates them using a dual-path attention model, focusing on the relationship between spliced and unspliced regions. DoveNet [7] treats image harmonization as a domain translation task. CDT-Net [6] combines pixel-to-pixel and RGB-to-RGB transformations for high-resolution image harmonization. [27] introduced the concept of style from the background image, treating the harmonization task as a style transfer problem. They proposed a novel Region-Aware Instance Normalization (RAIN) method, which extracts style information solely from the background features and applies it to the foreground of the image harmonization task.However, when the task extends to shadow generation, these methods do not scale well to handle both tasks simultaneously.

### 2.2 Shadow Generation

Previous work on shadow generation can be categorized into two main approaches: rendering-based methods and image-to-image translation methods.

Rendering-based methods [21, 24] relies on a clear understanding of lighting, reflectance, material properties, and scene geometry to generate shadows for inserted virtual objects using rendering techniques. However, such knowledge is often unavailable or impractical for applications in real-world scenarios. Image-to-image translation methods predominantly employ deep learning techniques, characterized by encoder-decoder architectures. By training on paired images, including images with shadows and those without, these methods directly learn the mapping from shadow-free images to shadowed images from the input data. Importantly, this approach typically eliminates the need for explicit knowledge about lighting, reflectance, material properties, and scene geometry. The ARShadowGAN [28] model introduces an attention-guided network capable of directly modeling the mapping relationship between the shadows of foreground objects and their corresponding real environments, accompanied by the release of the Shadow-AR dataset. SGRNet [17] promotes comprehensive information interaction between the foreground and background. It initially predicts a mask for shadow regions and subsequently forecasts shadow parameters to fill these regions. Additionally, a new shadow training dataset, DESOBA, is introduced. ShadowGAN [55] combines both global conditional discriminators and local conditional discriminators to generate shadows for inserted 3D foreground objects without relying on background lighting information.Shadow generation is

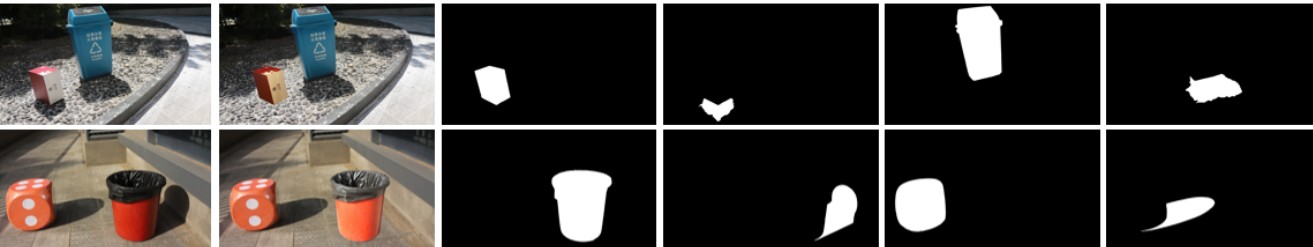

Figure 2: The display of the data pairs of the dataset we created, from left to right: ground truth, composite image, foreground object mask, shadow mask of foreground object, background object mask, shadow mask of the background object. We captured foreground and background separately under different lighting environments, paying attention to collecting various scenes, ground surfaces, and shadow casting situations.

associated with the foreground objects, but it targets different areas than image harmonization. We aim to combine both aspects using a single network framework.

## 3 PROPOSED METHOD

Given a composite image $I_c$, a binary mask $M_{bs}$ representing the background object-shadow pair, and a binary mask $M_f$ indicating the foreground object, our goal is to obtain an image $\tilde{I}$ that harmonizes the foreground object and produces reliable shadows under background illumination conditions.

As illustrated in Figure 4, our method consists of two stages: foreground object shadow mask prediction stage and shadow-harmonization generation stage. It mainly comprises four components: foreground object shadow generator $G_{fs}$, foreground content encoder $E(\cdot)$, adaptive encoder $E_a$, foreground content enhancement module $FCEM$, and shadow-harmonization generator $G(\cdot)$ based on a stable diffusion model.

The network workflow is as follows: Firstly, we use the background object-shadow data pair as reference to predict the shadow mask of the foreground object, identifying the approximate location for shadow generation. Then, we input the synthesized image into the shadow-harmonization generator to produce the final result. Simultaneously, we use the foreground content encoder to extract the foreground content embedding, inputting it into the foreground content enhancement module to constrain and complete foreground texture details. The adaptive encoder transfers the background style to the foreground region, providing additional generation guidance for the harmonization-shadow generator.

### 3.1 Harmony-Shadow Generator

Recently, diffusion models have shown remarkable performance in many fields: image generation [16, 45], text-to-image generation [39], image translation [25], image inpainting [32, 41], and image editing [12, 34]. The backbone of our Harmony-Shadow Generator is built upon a Stable Diffusion (SD) [39] model.

SD is a latent diffusion model that undergoes a two-stage pre-training process, involving an autoencoder and a denoising U-Net [16]. In the first stage, the SD model trains an autoencoder: the encoder $\mathcal{E}$ converts the images $I$ into a latent representation $z'_0 = \mathcal{E}(I)$, and then the decoder $\mathcal{D}$ reconstructs the images, resulting in

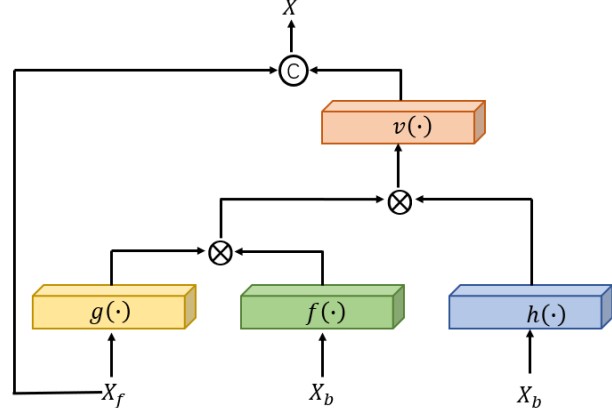

Figure 3: Overview of Cross-Attention Integration (Cross-Attention Integration) layer [17]. The $g, f, h, v$ shown in the figure represent $1 \times 1$ convolution, $\otimes$ represents matrix multiplication.

$\hat{I} = \mathcal{D}(z'_0)$. In the second stage, the autoencoder's parameters are fixed, and SD introduces noise to the latent space representation $z'_0$ over T steps to generate $z'_t$. This process involves the creation of a denoising U-Net $\epsilon_\theta$, which is trained using a latent denoising loss

$$\mathcal{L}_{LDM} := \mathbb{E}_{z'_0, y, \epsilon \sim \mathcal{N}(0,1), t} \left[ \| \epsilon - \epsilon_{\theta_1} (z'_t, t, \tau_{\theta_2}(y)) \|^2_2 \right], \qquad (1)$$

Here $\epsilon$ is the noise added to the latent space feature $z'_0$ at each noise step, $\epsilon_{\theta_1}$ is the denoising U-Net that predicts the noise $\epsilon$ at the current step $t$, and y represents additional conditions (e.g. text, mask, etc.), $\tau_{\theta_2}$ is instead a domain-specific encoder that projects y to an intermediate representation.

In this work, we add conditional information using an adaptive encoder similar to that of composite images with foreground masks. During the inference process, noise is first added to $z'_0$ to generate $z'_T$, and then $z'_T$ is used as $z_T$, which is the initial input of $\epsilon_{\theta_1}$. Then iteratively use $\epsilon_{\theta_1}$ to estimate the noise at each denoising step $t$, thereby gradually refining the latent map $z_T$, and ultimately become a clean latent feature $z_0$. Finally, the clean latent features $z_0$ are fed to the decoder $\mathcal{D}$ to generate images.

Figure 4: Overview of our CFDiffusion. It consists of two stages: foreground object shadow mask prediction stage and shadow-harmonization generation stage. It mainly comprises four components: foreground object shadow generator $G_{fs}$, foreground content encoder $E(\cdot)$, adaptive encoder $E_a$, foreground content enhancement module $FCEM$, and shadow-harmonization generator $G(\cdot)$. $E_f$ and $E_b$ respectively represent the foreground encoder and background encoder of the foreground object shadow mask generator.

## 3.2 Foreground shadow mask generator

Inspired by [17], we apply a shadow mask predictor to generate foreground object shadow predictor. First, we predict the foreground shadow mask $\widetilde{M}_{fs}$ through the foreground shadow mask generator $G_s$. $G_s$ consists of an encoder and a decoder $D$, and the encoder is divided into a foreground encoder $E_f$ and a background encoder $E_b$. We believe that the background object-shadow pair contains clues that are beneficial for inferring the foreground shadow area. In order to generate the shadow mask of the foreground object, we take the concatenation of the composite image $I_c$ and the background object-shadow mask $M_{bs}$ as the input of the background encoder $E_b$, and generate the background feature map $X_b$. At the same time, the concatenation of the composite image $I_c$ and the foreground object mask $M_f$ is used as the input of the foreground encoder $E_f$ to obtain the foreground feature map $X_f$. The process is summarized as follows:

$$X_b = E_b\left(M_{bs},\ I_c\right), \tag{2}$$

$$X_f = E_f\left(M_f,\ I_c\right). \tag{3}$$

Following [17], we use a Cross-Attention Integration (CAI) [17, 49, 50, 54] layer to help the foreground feature map notice the relevant lighting information of the background feature map. As the picture 3 shows, the input of the CAI layer consists of $X_f$ and $X_b$, which are outputs of foreground encoder $E_f$ and a background encoder $E_b$, and the output feature map is denoted as $X$. Then $X$ is fed into the decoder $D$ to obtain the mask of the foreground object shadow. Subsequently, we add it to the foreground object mask $M_f$ to obtain the foreground object-shadow mask $\widetilde{M}_{fs}$, which serves as one of the inputs for the subsequent shadow-harmonization generator. The process is summarized as follows:

$$\widetilde{M}_{fs} = D(X) + M_f. \tag{4}$$

## 3.3 Foreground Encoder

Following [53], in order to further enhance the detailed texture of the foreground generation, we employ the pre-trained model $ViT-L/14$ from CLIP [38] as the foreground image encoder $E$. Initially, we extract the foreground object region from the synthesized image $I_c$ using the foreground object mask $M_f$, which is then inputted into the foreground image encoder $E$ to extract the local content embedding of the foreground $E_l$. This process can be represented as follows:

$$E_l = E(I_f). \tag{5}$$

The intermediate layer of the CLIP encoder outputs 256 patch tokens containing local details. We extract the information of these patch tokens and integrate these foreground content embeddings into the Foreground content enhancement module (FCEM) of the denoising U-Net model to help us control the generation of foreground content details. The specific details of the FCEM module is located in Section 3.4.

## 3.4 Foreground content enhancement module(FCEM)

Following [53], we utilize a foreground content enhancement module to embed foreground content into the intermediate features of the diffusion model, thereby constraining the stable diffusion model for foreground appearance generation and promoting the composite generation of foreground appearance with high fidelity.

Our foreground content enhancement module is built upon the publicly released v1-4 SD model. To identify foreground regions that need to be constrained, we append the binary foreground object-shadow mask $\widetilde{M}_{fs}$ to the model input. To achieve this, in the first convolutional layer of U-Net, we attach two additional input channels to respectively contain the foreground object mask $M_{fs}$

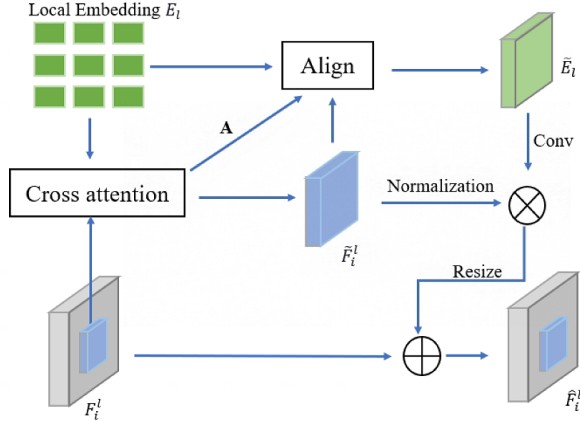

**A** : Attention map    $\otimes \oplus$ : Element-wise multipliaction/addition

**Figure 5: Overview of FCEM module. A in the figure is the attention map output by the cross attention part.** $F_i \in \mathbb{R}^{h_i \times w_i \times c_i}$ **is feature map of i-th transformer block**

and the predicted foreground object-shadow mask $\widetilde{M}_{fs}$. Eventually, the input images are uniformly resized to a resolution of $256 \times 256$ .

Denoising U-Net of SD consists of a series of basic blocks, each block includes a residual block and a transformer block. The transformer block consists of a self-attention module, a cross-attention module, and a feedforward network.

As illustrated in Figure 5. We record the features from the $i - th$ transformer block as $F_i \in \mathbb{R}^{h_i \times w_i \times c_i}$, where $h_i, w_i, c_i$ represent its height, width, and channel dimensions respectively. We first use the foreground local feature $F_i^l$ intercepted by the foreground object mask ($M_f$) resized to $h_i, w_i, c_i$. The feature map will be flattened to $\bar{F}_i^l \in \mathbb{R}^{N \times c_i}$, and then passed through cross attention together with the foreground local embedding $E_l$, and then we get an attention map $A$ and refined foreground local feature map $\widetilde{F}_i^l$. Then align the foreground local embedding $E_l$ with $\widetilde{F}_i^l$ to obtain the aligned foreground embedding map $\widetilde{E}_l$. Further use $\widetilde{E}_l$ to modulate $\widetilde{F}_i^l$. $\widetilde{E}_l$ is passed through a convolutional layer of $3 \times 3$ to obtain the spatial awareness modulation weight, and the modulation is normalized The transformed $\widetilde{F}_i^l$ is as follows:

$$\hat{F}_i^l = norm(\widetilde{F}_i^l) \bullet conv(\widetilde{E}_l). \tag{6}$$

Finally, after resizing $\hat{F}_i^l$, it is added to the foreground object region of $F_i$. The output of the Foreground Content Enhancement Module (FCEM) is then delivered as the enhanced foreground content features $\widetilde{F}_i$ to the next residual block.

## 3.5 Adaptive Encoder

Following [31, 35], we adopt an adaptive encoder, which is a lightweight model that can align the internal knowledge in the SD model with external control signals. Through this adaptive encoder, we can achieve rich control effects on the color and structure of the SD generation results.

The adaptive encoder takes into account encoding additional conditions and provides multi-step guidance for denoising U-Net in the denoising step. Previous adaptive encoder implementations focused more on coarse structures (e.g., sketches, poses, semantic masks) and exploited textual conditions to indicate additional requirements (e.g., style or context). Different from previous work, we abandon the text CLIP model, splice the composite image $I_c$ and the foreground mask $M_f$, and use a lightweight adaptive encoder to encode while retaining content details and extracting background styles. The structure of the adaptive encoder includes four feature extraction blocks and three DownSample (DS) blocks.

First, the input image will be resized to $64 \times 64$, and we name it $F_c^0$. Each feature extraction module (EM) includes a convolutional layer and two residual blocks. The generation process of $F_c^i, i \in 1, 2, 3, 4$ can be expressed as follows:

$$F_c^1 = EM_1\left(F_c^0\right), \tag{7}$$

$$F_c^i = EM_i\left(DS(F_c^{i-1})\right). \tag{8}$$

The resolutions of $F_c^1, F_c^2, F_c^3$, and $F_c^4$ are $64 \times 64, 32 \times 32, 16 \times 16, 8 \times 8$ respectively. Then we use foreground mask $M_f$ to separate the foreground features $F_{c,f}^i$ and background features $F_{c,b}^i$:

$$F_{c,f}^i = Flatten(F_c^i \circ M_f), \tag{9}$$

$$F_{c,b}^i = Flatten(F_c^i(1 - M_f)). \tag{10}$$

Among them, $M_f$ represents the foreground object mask scaled to the corresponding $F_c^i$ size, $\circ$ represents element-wise product, and $Flatten(\cdot)$ represents expanding a 2D feature map into a 1D feature feature sequence.

We use a transformer layer to extract and transfer the style of the background to the foreground area to achieve harmonious processing of the foreground. In addition, the parts of the background area that are related to the foreground can provide more references when harmonizing the foreground, so they are very important. We will also pay more attention to the areas that are more related to the background and foreground. $F_{c,f}^i$ is used as query, $F_{c,b}^i$ is used as keys/values, and the final background stylized foreground feature $\hat{F}_{c,f}^i$ can be expressed as:

$$\hat{F}_{c,f}^i = Transformer(F_{c,f}^i, F_{c,b}^i, F_{c,b}^i). \tag{11}$$

## 3.6 Traning Losses and Details

Our total loss function $L_{total}$ consists of the standard noise loss $\mathcal{L}_{LDM}$ of the diffusion model and a reconstruction loss $L_{rec}$, Therefore, the final loss function of our CFDiffusion is:

$$L_{total} = \lambda_1 L_{rec} + \lambda_2 \mathcal{L}_{LDM} + \lambda_3 L_{fs}, \tag{12}$$

where $\lambda_1, \lambda_2, \lambda_3$ are hyper-parameters which control the influence of terms.

**Noise Loss.** First, we adopt the standard noise loss of the diffusion model, aiming to reconstruct the image features in the latent space, shown as Equation (13):

$$\mathcal{L}_{LDM}: = \mathbb{E}_{z_0', y, \epsilon \ \mathcal{N}(0,1), t}[\| \epsilon - \epsilon_{\theta_1}(z_t', t, \tau_{\theta_2}(y)) \|_2^2]. \tag{13}$$

**Reconstruction loss.** It is a classical $L_1$ loss between the generator output image $\hat{I}$ and real ground-truth image $I$, to further constrain

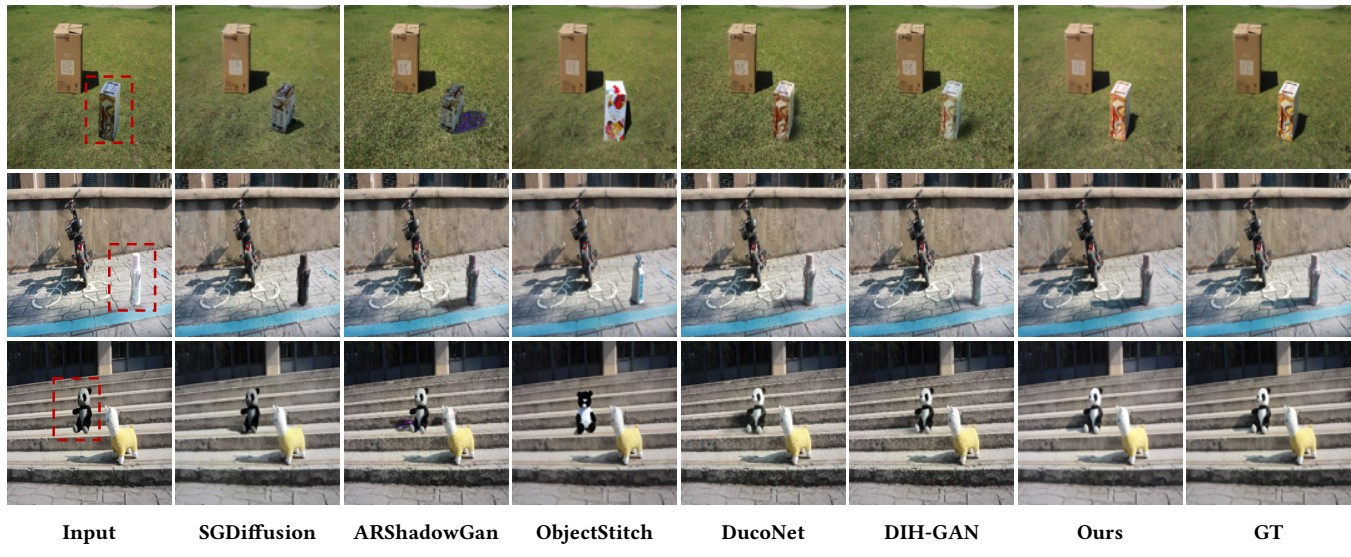

| Input | SGDiffusion | ARShadowGan | ObjectStitch | DucoNet | DIH-GAN | Ours | GT |

Figure 6: The comparison between our method and two image harmonization methods: DucoNet [48] and DIH-GAN [2], three shadow generation methods [28, 29, 46] on our dataset. It can be clearly seen that our CFDiffusion has achieved the best results in both real shadow generation and image harmonization.

the generated image towards the ground truth, which is expressed as:

$$L_{rec} = \| \widetilde{I} - I \|_1 . \tag{14}$$

**MSE Loss.** Additionally, we compute the loss for the foreground mask prediction module using the following method:

$$L_{fs} = \| M_{fs} - \widetilde{M}_{fs} \|_2^2 . \tag{15}$$

## 4 EXPERIMENTS

To verify the superiority of our proposed CFDiffusion, we compare CFDiffusion with state-of-the-arts on the real-world iHarmony4 [7] dataset and our proposed dataset, and provide assessments both quantitatively and qualitatively.

### 4.1 Experimental Settings

The proposed method is implemented using PyTorch, which is trained using one NVIDIA RTX 3090 GPU. All images are resized to $256 \times 256$ for training and testing. We adopt adam optimizer with the momentum as (0.9,0.999), and the learning rate initialized as 0.00003. Following [42], we use the Kaiming initialization technique [14] to initialize the weights of the proposed model and use a 0.9999 Exponetial Moving Average(EMA) for all our experiments. We used 1000 diffusion steps T and noise schedule $\beta_t$ linearly increasing from 0.0001 to 0.002 for training, and 25 steps for inference. After a few trials, we set $\lambda_2 = \lambda_3 = 10$, $\lambda_1 = 1$ by observing the grenerated images. The training epoch is set as 1500.

**Compared methods.** We compared our model with five deep learning-based methods from the related fields: two image harmonization methods including DucoNet [48], DIH-GAN [2], three shadow generation methods include ARShadowGAN [28], SGDiffusion [29], and ObjectStitch [46]. Among them, SGDiffusion is the latest method using a diffusion model for shadow generation tasks, while ObjectStitch is the newest image composition method

that addresses both image harmonization and shadow generation tasks. DucoNet for image harmonization based on dual color spaces. ARShadowGAN makes full use of the background information to guide the shadow generation of foreground objects. We train and test all these methods based on our dataset. For detailed information about our IH-SG dataset, please refer to the remainder of this section.

In addition, we also tested our image harmonization capabilities on the iHarmony4 dataset, and compared the three image harmonization methods of CDT-Net [6], Harmonizer [23] and DucoNet [48] to further prove the superiority of our CFDiffusion. CDT-Net coherently combines pixel-to-pixel conversion and RGB-to-RGB conversion in an end-to-end network.

**Evaluation metrics.** We use four metrics to evaluate the image illumination harmonization results, which are Relative Mean Square Error (RMSE), Structural Similarity Index Measure (SSIM), foreground Mean Square Error (fMSE), foreground Structural Similarity Index Measure (fSSIM). Generally, the smaller RMSE and fMSE, and the larger fSSIM and SSIM indicate the better image illumination harmonization results.

**Dataset.** To better train our model, we have constructed a dataset called IH-SG that can address both of image harmonization and shadow generation concurrently. Each data pair we construct includes: composite images, ground truth images, foreground object masks, foreground object shadow masks, background object masks and background object shadow masks.

In total, our dataset comprises over 1000 outdoor scenes and more than 10000 data pairs. We also captured numerous shadow scenes under complex conditions, such as shadows cast on walls and steps, to enrich the shadow data samples, making our dataset more realistic and diverse. Our data pairs are illustrated in the Figure 2.

## 4.2 Comparison with State-of-the-Arts

**Experiments on our dataset.** The quantitative comparison results of different methods on our testing set are summarized in Table 1. Apparently, our method CFDiffusion achieves the better quantitative results than other state-of-the-art methods on all four metrics. On the one hand, this is due to the powerful image generation capability of the diffusion model. On the other hand, our FCEM and adaptive encoder make full use of the foreground and background information, proving the superiority of our CFDiffusion.

**Experiments on iHarmony4.** The iHarmony4 dataset is one of the most popular large-scale datasets in the field of image harmonization, covering a variety of scenes and foreground objects. Therefore, we compared our method with several image harmonization techniques on the iHarmony4 dataset, and the quantitative comparison results are shown in Table 3. Our proposed approach also achieved the best results.

**Experiments on DESOBAv2.** Recently, Liu et al. [29] extended the DESOBA [17] dataset to DESOBAv2, which has become the latest shadow generation dataset. To validate the generalization ability of our model, we conducted experiments on the DESOBAv2 dataset by applying slight perturbations to foreground objects, and the results are shown in Figure 13.

## 4.3 Ablation Study

In order to verify the effectiveness of each component of our method, we conduct ablation studies by modifying the CFDiffusion architecture. Specifically, we set the following variants:

To verify the crucial roles played by our FCEM module and adaptive encoder module in the overall model, we set up several variants. Firstly, we chose the original diffusion model as the baseline, which is referred to as "*baseline*" in Table 2. To demonstrate the pivotal roles of the FCEM module and adaptive encoder module in the entire model, we removed these two modules separately from the complete model, which are referred to as "*w/o FCEM*" and "*w/o adaptive encoder*" in Table 2. Finally, we compared these variants with the full model "*Ours (full model)*" for comprehensive analysis, and some results are shown in Figure 12.

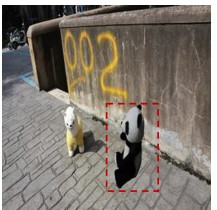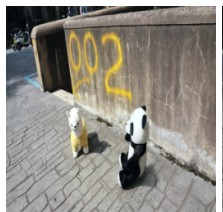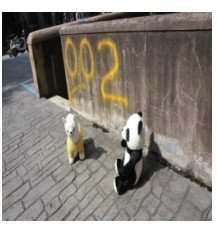

    **Composite image**        **Result**          **GT**

**Figure 10: In order to simulate scenarios where there are no background objects available as reference, we only utilize the foreground encoder module during the foreground shadow mask prediction stage, hence not referring to background shadow information. Consequently, the inferred shadow shapes will deviate significantly from the ground truth.**

**Table 1: Results of quantitative comparison on our testing set. "↑" indicates the higher the better, and "↓" indicates the lower the better. The best results are marked in bold.**

| Method | RMSE ↓ | SSIM ↑ | fMSE ↓ | fSSIM ↑ |
|---|---|---|---|---|
| SGDiffusion [29] | 8.591 | 0.825 | 875.882 | 0.809 |
| ARShadowGAN [28] | 9.164 | 0.817 | 942.154 | 0.816 |
| ObjectStitch [46] | 9.357 | 0.773 | 1145.116 | 0.773 |
| DucoNet [48] | 7.346 | 0.861 | 454.213 | 0.915 |
| DIH-GAN [2] | 6.145 | 0.847 | 567.311 | 0.894 |
| Ours | **5.582** | **0.917** | **367.919** | **0.937** |

We trained these variants using the same training data and quantitatively evaluated their impact on the test results. The evaluation results are presented in Table 2. From the table, we can observe that: after introducing guided supervision with FCEM, the model's quantitative performance has made significant strides, sufficiently demonstrating the strong guiding role of the FCEM module in capturing texture details of foreground objects. Moreover, with the inclusion of the adaptive encoder, the model's performance has also noticeably improved compared to the original diffusion model, confirming its guiding role in the harmonization generation of foreground objects.

With the simultaneous introduction of both the FCEM and adaptive encoder modules, our full model achieved the best performance, demonstrating the effectiveness of our approach. Additionally, incorporating the FCEM and adaptive encoder modules into the original diffusion model significantly improves performance.

**Table 2: Ablation study of FCEM and adaptive encoder. The best results are marked in bold.**

| Method | RMSE ↓ | SSIM ↑ | fMSE ↓ | fSSIM ↑ |
|---|---|---|---|---|
| baseline | 11.192 | 0.687 | 543.417 | 0.675 |
| w/o FCEM | 7.264 | 0.746 | 398.542 | 0.785 |
| w/o adaptive encoder | 8.437 | 0.727 | 485.534 | 0.841 |
| Ours | **5.582** | **0.917** | **367.919** | **0.937** |

## 4.4 Limitations

Our CFDiffusion still has the following limitations: (1) As shown in Figure 10, for scenarios lacking background object references or involving complex situations where shadows are cast onto intersecting planes, our model struggles to generate shadows effectively. (2) Due to computational costs and processing speed limitations, our method is currently not applicable to real-time video lighting harmonization, which is also one of our future directions for improvement.

## 4.5 Conclusion and Future Work

In this paper, we proposed an image composite method based on the diffusion model, which focuses on harmonizing the illumination inconsistency between foreground objects and background, while generating realistic shadows. We employed an adaptive encoder to extract style features from the background to guide the

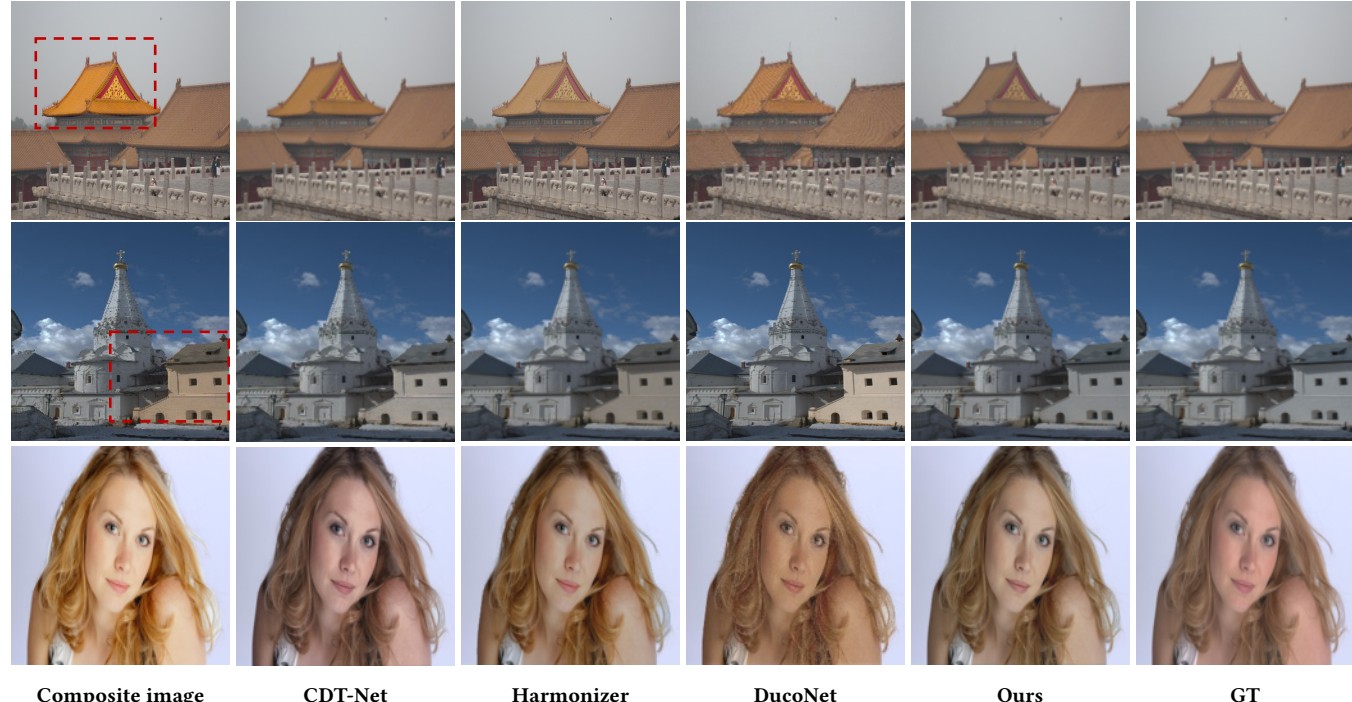

| Composite image | CDT-Net | Harmonizer | DucoNet | Ours | GT |
|---|---|---|---|---|---|

Figure 11: The visual results of harmonization experiments on iHarmony4 [7]. It can be seen that our results are closest to the Ground Truth.

Table 3: Results of quantitative comparison on iHarmony4. "↑" indicates the higher the better, and "↓" indicates the lower the better. The best results are marked in bold.

| Method | RMSE ↓ | SSIM ↑ | fMSE ↓ | fSSIM ↑ |
|---|---|---|---|---|
| CDT-Net [6] | 6.847 | 0.804 | 379.187 | 0.858 |
| Harmonizer [23] | 6.308 | 0.854 | 410.847 | 0.821 |
| DucoNet [48] | 6.152 | 0.876 | 365.236 | 0.915 |
| Ours | **5.582** | **0.917** | **367.919** | **0.937** |

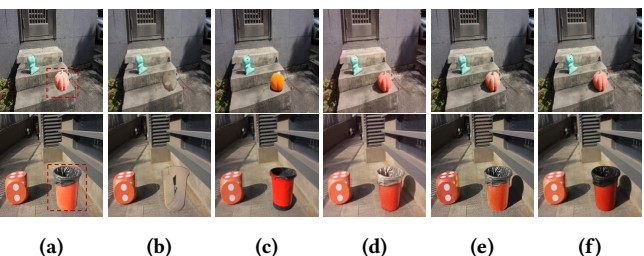

(a)       (b)       (c)       (d)       (e)       (f)

Figure 12: Ablation experiment results. From (a) to (f), they are respectively the composite images, w/o FCEM, w/o adaptive encoder, our complete method, and GT.

diffusion model in better harmonizing the foreground. Specifically, we introduced a FCEM module to further improve the ability to preserve details of foreground content. Finally, we have conducted

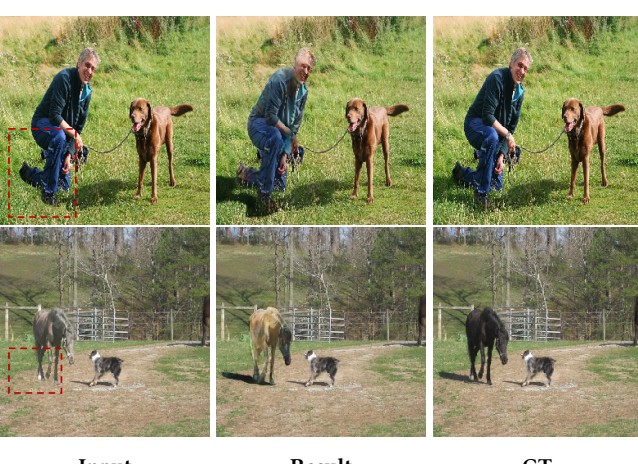

| Input | Result | GT |
|---|---|---|

Figure 13: The experimental results on DESOBAv2 dataset. The area enclosed by the red box is where the shadow is expected to be generated.

experiments on our proposed IH-SG dataset, as well as the popular DESOBAv2 dataset and iHarmony4 dataset, demonstrating that our method achieves significant improvements. In the future, we will expand CFDiffusion to adapt to real-time video lighting harmonization and shadow generation.

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
