# OpenReview forum: "CFDiffusion: Controllable Foreground Relighting in Image Compositing via Diffusion Model"
_acmmm.org/ACMMM/2024/Conference — MM2024 Poster_

### Official Review · Reviewer_XYGY · 2024-05-13

**Rating:** 2
**Confidence:** 3

**Summary:**

This article presents a decomposition of the image synthesis task into two subtasks: shadow generation and image harmonization. Subsequently, the article proposes an image composite method based on the diffusion model. This method seamlessly integrates foreground objects into a given background scene and relights foreground objects while generating shadows to better blend them into the background and eliminate any sense of incongruity between them.

**Strengths:**

1.The experimental results indicate that the proposed approach can effectively address the issue of insufficient foreground fidelity in image synthesis.

2.This work constructs a dataset that can address both of image harmonization and shadow generation concurrently.

**Limitations:**

About the method
The article's primary deficiency is the lack of innovation. It appears that there is no difference between the foreground shadow mask generator module in this paper and the shadow mask prediction module in [1], and the foreground encoder module and FCEM in this paper are no different from the module of the same name in [2]. The whole paper is basically composed by adding an auto-encoder module on the basis of [1] and [2].

About the experiments
The work in this article contains two subtasks, image coordination and shadow generation, but the experimental part of it is not particularly well described.

1.SGDiffusion is just a shadow generation method, whereas Table 1 represents image coordination work and how you use it to perform image coordination work on your dataset. Similarly, the other methods compared are either shadow generation or image coordination single-task solutions, and the article lacks content on how that component is a fair comparison。

2.The DESOBAv2 dataset is a shadow generation dataset. Are the experiments on this dataset simply to assess the ability of shadow generation? Additionally, the article lacks objective metric results on that dataset. Would the results be consistent with those of [1]? (How does the article's approach to shadow generation differ from [1]?)

3.The caption of Figure 12 refers to five images, however, there are six images within the picture. It is therefore necessary to re-add the description.

About the writing

1.Row 506 seems to be missing a full stop.

2.Why does the formula 11 input have two identical elements？
The author should carefully review their manuscript for further writing errors.

[1]Yan Hong, Li Niu, and Jianfu Zhang. 2022. Shadow Generation for Composite Image in Real-World Scenes.
[2]Bo Zhang, Yuxuan Duan, Jun Lan, Yan Hong, Huijia Zhu, Weiqiang Wang, and Li Niu. 2023. Controlcom: Controllable image composition using diffusion model.

**Suitability:**

2

---

### Official Review · Reviewer_W7cx · 2024-05-22

**Rating:** 4
**Confidence:** 3

**Summary:**

The paper presents CFDiffusion, a unified model simultaneously handling image harmonization and shadow generation. After first employing a mask predictor to obtain a shadow mask, the paper proposes to use FCEM and an adaptive encoder to help with appearance preservation and harmonization.

**Strengths:**

- The motivation of combining shadow synthesis and harmonization is reasonable, as these effects share some common information from the global background area.
- The paper shows extensive quantitative and visual results, which demonstrate the advantages of the proposed model.

**Limitations:**

- Fig 2 is not mentioned until Sec 4.1. It would be better to switch the order of the figures.
- Fig 4 needs to be revised for easier understanding. It could be improved in the following ways: 1) some notations should be added to the figure, such as G_fs; 2) maybe use two boxes to illustrate the two stages; 3) the exact inputs should be visualized to show how they are processed by the encoders, e.g., what’s the input of E_a? Is it the masked composite image or the concatenation of I_c and M_f?
- Fig 5 is confusing. How is the “align” operation performed?
- An ablation on the reconstruction loss is missing. This would be beneficial to show whether identity preservation is contributed by FCEM or the reconstruction loss.
- (Minor) For the comparison with baselines, maybe also consider AnyDoor and ControlCom (ControlCom has a harmonization mode).
- How does the model perform in indoor scenes where there are soft shadows? Such examples are missing from the paper.
- In the third row of Fig 11, it seems the proposed model does not have an obvious advantage over the baselines?
- In Tab 3, DucoNet should be the one highlighted regarding the fMSE score.

**Suitability:**

3

---

### Official Review · Reviewer_cndV · 2024-05-24

**Rating:** 4
**Confidence:** 2

**Summary:**

This paper proposes CFDiffusion, a unified model that utilizes diffusion models to simultaneously tackle image harmonization and shadow generation tasks.  It includes two main stages: foreground object shadow mask prediction stage and shadow harmonization generation stage. This method has been evaluated with comprehensive experiments.

**Strengths:**

This paper is well-organized and straightforward. The experiments clearly demonstrate the advantages of the proposed pipeline, particularly showing the benefits of the adaptive encoder and the Foreground Content Enhancement Module (FCEM).

**Limitations:**

[1]. Please show the inference time between the different methods.

[2]. Are there any quantitative results comparisons on the DESOBAv2 dataset?

[3]. You can add more details about FCEM in part 3.4 to illustrate the Figure5.

**Suitability:**

2

---

### Official Review · Reviewer_Tho3 · 2024-05-25

**Rating:** 1
**Confidence:** 4

**Summary:**

This paper proposes CFDiffusion to simultaneously handle image harmonization and shadow generation. The experimental results on the iHarmony4 dataset and our IH-SG dataset demonstrate the superiority of CFDiffusion.

**Strengths:**

The results seems good.

**Limitations:**

1.	The proposed method is a mix of existing methods or models (SGRNet, diffusion model) and lacks novelty.
2.	The authors frequently mention the IH-SG dataset as their constructed dataset. For example, line 49 (our IH-SG), line 153 (our proposed IH-SG dataset), line 165 (our newly created dataset IH-SG). If the IH-SG dataset is the contribution, please add it to the paper. If the dataset is publicly available, I'm considering whether this breaks the principle of anonymity.
3.	The manuscript feels like being rushed. In the abstract, two datasets are claimed (line 49). In the conclusion section, three datasets are claimed (Line 923-924).

**Suitability:**

2

---

### Meta-Review · Area_Chair_kdAb · 2024-06-26

**Recommendation:** Accept (Poster)
**Confidence:** 5

**Metareview:**

This paper receives a mixture of reviews, two borderline accepts, one weak reject, and one reject. The reviewers giving borderline accepts acknowledge the motivation and effectiveness of the proposed method. The review of the reviewer giving reject is very short and not very informative. The concern of the other reviewer giving weak reject mainly lies in the novelty of the proposed method, which seems to be a combination of existing methods. I believe that the constructed dataset is a huge contribution to the community. The proposed method is also not a naive combination and has some interesting designs. The first stage provides useful prior information for the second stage, which alleviates the burden of diffusion model in the second stage. The foreground information is injected into denoising UNet through two ways to ensure the foreground quality. By weighing the advantages and drawbacks, I tend to accept this paper.